# How Physical Activity across the Lifespan Can Reduce the Impact of Bone Ageing: A Literature Review

**DOI:** 10.3390/ijerph17061862

**Published:** 2020-03-13

**Authors:** Maria Felicia Faienza, Giuseppe Lassandro, Mariangela Chiarito, Federica Valente, Loredana Ciaccia, Paola Giordano

**Affiliations:** 1Department of Biomedical Sciences and Human Oncology, Paediatric Section, University of Bari “A. Moro”, 70124 Bari, Italy; giuseppelassandro@live.com (G.L.); mariangelachiarito@gmail.com (M.C.); loredana.ciaccia89@gmail.com (L.C.); paola.giordano@uniba.it (P.G.); 2Section of Cardiovascular Diseases, Department of Emergency and Organ Transplantation, University of Bari, School of Medicine, 70124 Bari, Italy; valentefederica91@gmail.com

**Keywords:** physical activity, bone health, childhood, ageing

## Abstract

Bone remodeling is a lifelong process, due to the balanced activity of the osteoblasts (OBs), the bone-forming cells, and osteoclasts (OCs), the bone-resorbing cells. This equilibrium is mainly regulated by the WNT-ß-cathenin pathway and the RANK-RANKL/OPG system, respectively. Bone ageing is a process which normally occurs during life due to the imbalance between bone formation and bone resorption, potentially leading to osteoporosis. Bone loss associated with bone ageing is determined by oxidative stress, the result of the increasing production of reactive oxygen species (ROS). The promotion of physical exercise during growth increases the chances of accruing bone and delaying the onset of osteoporosis. Several studies demonstrate that physical exercise is associated with higher bone mineral density and lower fracture incidence, and the resulting bone mineral gain is maintained with ageing, despite a reduction of physical activity in adulthood. The benefits of exercise are widely recognized, thus physical activity is considered the best non-pharmacologic treatment for pathologies such as osteoporosis, obesity, diabetes and cardiovascular disease. We reviewed the physiological mechanisms which control bone remodeling, the effects of physical activity on bone health, and studies on the impact of exercise in reducing bone ageing.

## 1. Introduction

Bone remodeling is a dynamic process which occurs throughout life, to replace old and damaged bone with the new one [1,2]. It takes place in the “basic multicellular units” (BMUs) consisting of cluster of osteoclasts (OCs), the bone-resorbing cells, and osteoblasts (OBs), the bone forming cells, which work sequentially [3]. Bone modeling is responsible for the shape and mechanically induced adaption of bones, and OBs and OCs can act independently at distinct anatomical sites [1,3]. In healthy subjects, bone formation mainly occurs in the first two decades of life, until the achievement of peak bone mass. Thereafter, bone mass remains stable for approximately 20 years, until resorption begins to outweigh bone formation with subsequent age-related bone loss [1]. Sixty percent of the risk of osteoporosis depends on what happens in the first two decades of life, while the remaining 40% on what happens after [1].

In this review, we focus on physiological mechanisms which control bone remodeling, the effects of physical activity on bone health, and we update studies on the impact of exercise in reducing bone ageing. We performed a systematic literature search in PubMed and EMBASE, reviewed and selected articles, based on the following key words: ‘physical activity’, ‘bone health’, ‘childhood’, ‘ageing’.

## 2. Physiological Mechanisms of Bone Remodeling

Osteoblast differentiation is controlled by the master transcription factor RUNX2 (runt-related transcription factor 2), and is characterized by four stages: the preosteoblast, osteoblast, osteocyte and bone-lining cell. These cells contribute differently to bone remodeling, according to their differentiation stage. In particular, immature OBs direct osteoclastogenesis, whereas only mature OBs have the ability to produce mineralized tissue [2,3]. The canonical Wnt/β-catenin pathway is critical for bone development. When Wnt signaling is activated, Wnt proteins bind to Frizzled receptor and low-density lipoprotein receptor-related proteins five and six (LRP5, LRP6). The consequent hypophosphorylated state of β-catenin prevents its degradation, and it results in the upregulation of transcription factors crucial for osteoblast differentiation [4,5]. The Wnt signal is modulated by different antagonists, including sclerostin (SOST), Dickkopf-1 (Dkk-1), and secreted frizzled-related proteins (sFRP), which inhibit osteoblastogenesis [5].

Osteoclastogenesis is under the control of two factors: the macrophage-colony stimulating factor (M-CSF), and the receptor activator of nuclear factor kappa-B ligand (RANKL). The binding of these factors to their respective receptors, c-fms (colony-stimulating factor-1 receptor) and RANK (receptor activator of nuclear factor kappa-B), on osteoclast precursors, starts osteoclastogenesis. The RANKL-RANK binding can be antagonized by osteoprotegerin (OPG), a soluble decoy receptor secreted by OBs and bone marrow stromal cells, which binds to RANK and prevents the osteoclastogenic effect of RANKL [1]. RANKL and OPG are also produced by activated T-cells, which represent a key paracrine link between bone metabolism and the immune system [6].

Under physiological conditions, a balance between bone resorption and bone formation ensures the strength and integrity of the human skeleton. Several pediatric disorders can lead to an altered peak bone mass (PBM) and therefore bone loss, thus resulting in an increased risk for osteoporosis and fractures [7]. In particular, literature data have demonstrated an involvement of RANKL, OPG, sclerostin and DKK-1, both in inherited and acquired pediatric diseases [8,9,10,11,12].

Among the risk factors for osteoporosis, it is possible to identify modifiable factors, including diet and lifestyle factors, and non-modifiable factors such as gender, age, genetic factors, history of prior fractures, diseases and pharmacological treatments [13,14]. The modifiable factors, such as a balanced diet and exercise, have an important role already in childhood. In particular, regular physical activity has a key role in bone strengthening, not only in healthy children, but also in those suffering from chronic diseases [15,16,17].

## 3. Physical Activity and Bone Health: The Metabolic, Inflammatory and Immune Response

Bone metabolism is significantly affected by exercise, resulting in an adaptation of bones in terms of shape, mass, and strength to the mechanical loading. During physical activity, bone tissue deforms, and the mechanosensors located through the cells, such as ion channels and integrins, change their original conformation triggering several signals, including calcium, mitogen-activated protein kinase (MAPK), Wnt, and RhoA/ROCK pathways [18]. In particular, the mechanical loading activates the Wnt/β-catenin signaling pathway, either by direct stimulation of the bone transcription factor RUNX2, or by a cross-talking with parathyroid hormone (PTH) or morphogenetic proteins (BMPs) signaling pathways [19]. By regulating the OPG and RANKL expression in OBs, Wnt signaling also downregulates osteoclastogenesis and osteoclast activity [20]. Furthermore, loading activates a molecular response that inhibits sclerostin and allows the activation of Wnt signaling, thereby osteoblastogenesis and bone formation. Finally, exercise shifts the adipogenic-to-osteogenic balance toward the osteoblast formation [21].

Exercise duration, type, and intensity are the factors which determine the exercise-induced inflammatory response. Physical activity activates an inflammatory cascade involving cells of the innate and adaptive immunity, as cytokines, and mediators of inflammation, as myokines and adipokines, which create an environment adapt for the recovery, regeneration, and adaptation of bone (Figure 1) [22]. Among the myokines, irisin is involved in both glucose and bone homeostasis [23]. It is secreted by skeletal muscle during physical activity, and it induces osteogenesis at the bone–muscle interface [24]. Regular and moderate physical activity seems to maintain higher irisin levels in normal-weight adolescents compared with their sedentary counterparts [25]. Irisin appears positively correlated with bone mineral density (BMD) and bone strength in young athletes and in soccer players, supporting the idea that irisin could have a protective role on bone health [23]. In healthy children, irisin levels are positively correlated with bone mineral status and circulating osteocalcin [26]. High irisin levels have also been observed in children and adolescents with type 1 diabetes mellitus (T1DM), as they are associated with a better metabolic control and improved bone mass [27]. These evidences suggest that irisin might be considered as one of the bone formation markers during childhood.

Physical activity is also able to modify bone functions by modulating the immune system, thus, in the last 15 years, the “osteoimmunology” has become central in studying the metabolic diseases of bone [28]. Furthermore, primary and secondary osteoporosis have been treated with fully human monoclonal antibodies which act as inhibitors of the osteoimmunological signaling pathway, the RANK/RANKL [29]. Exercise also activates the inflammasome complexes, and increases IL-6 levels, which in turn plays an anti-inflammatory effect by inhibiting the release of pro-inflammatory cytokines (TNFα and IL-1β), and triggering the release of IL-10, a potent anti-inflammatory molecule (Figure 1) [18,30,31]. In particular, acute physical activity increases pro-inflammatory cytokines, whereas regular exercise results in an enhancement of anti-inflammatory molecules [30,32].

## 4. Impact of Physical Activity in Reducing Bone Ageing

Bone ageing is a process normally occurring over time, which leads to imbalance between osteoclast resorption and osteoblast bone formation. Genetic factors and epigenetic modifications triggered by lifestyle affect this process [33]. The bone loss associated with bone ageing is determined by oxidative stress that negatively affects the signaling pathways implicated in bone cell survival and osteogenesis [34]. The production of mitochondrial superoxide anion in elderly osteocytes increases bone resorption [35], while the reactive oxygen species (ROS) reduces the β-catenin signaling, with activation of peroxisome proliferator-activated receptor (PPAR) γ supporting adipogenesis at the expense of osteoblastogenesis [36]. Moreover, the activation of oxidative defense Forkhead box O (FOXO) transcription signaling involved in ageing and longevity triggers the apoptosis of OBs and osteocytes (Figure 2) [37]. On the contrary, physical activity positively affects bone metabolism via different mechanisms: 1. activation of inflammatory cascade involving cells of the innate and adaptive immunity and mediators of inflammation; 2. triggering a metabolic response due to the increase of IL-6 by skeletal muscle; 3. stimulation of the Wnt signaling pathway (Figure 1).

Mechanical loading also reduces the adipogenic differentiation of mesenchymal stem cells by rescuing ß-catenin-FOXO mediated transcription [38]. Furthermore, exercise decreases osteocyte apoptosis, as demonstrated in ovariectomized mice [39], and preserving telomere from progressive shortening [40].

### Physical Activity during the Lifespan

The promotion of physical exercise during growth increases the chances of accruing bone and delaying the onset of osteoporosis. To date, there are several evidence that physical exercise is associated with higher BMD and lower fracture incidence, and the resulting bone mineral gain is maintained with ageing, despite a reduction of physical activity in adulthood [41]. However, while during growth physical exercise increases the PBM, during adulthood weight-bearing exercises should be performed to maintain bone mass and increase bone strength [42]. Indeed, although it is known that exercise may improve BMD in postmenopausal women and adult men, the type of activity, its intensity, duration and frequency, are still unclear. Further studies are needed to determine the precise training protocol, the dose-response relationship and whether any associations persist into adulthood [43,44].

Infancy, childhood and adolescence are critical periods for the development of the skeleton. During these periods, mechanical load is one of the best stimuli to enhance bone mass and structural skeletal adaptations, both contributing to bone strength [41]. It is known that both genetics and physical activity contribute to BMD. However, it is unknown if the benefits of exercise on childhood bone accretion are influenced by genetic factors. Mitchell et al. observed the beneficial effects of exercise in children genetically predisposed to a lower BMD in adulthood [45]. However, the timing of beginning of physical activity would seem to be important. Indeed, the age at which children start walking might influence their bone strength later in life [46]. Modifications in bone structure and strength related to exercise are most often observed in prepubertal and peripubertal age, and although some changes are sex related, physical activity is generally associated with improved bone strength, both in boys and girls [47,48]. It is known that the time period just prior to puberty represents a “window of opportunity”, when the skeleton is most sensitive to mechanical loading [49]. However, it is possible that the window of opportunity may occur at different maturity levels for males and females, and may be shorter for females than males [49]. Extending the “window of opportunity” concept to include an earlier time period has been suggested. In fact, the increase of physical activity since infancy and childhood may improve compliance to regular exercise. Systematic sporting activity during childhood is associated with increased BMD when comparing active to inactive groups. However, the degree to which different sport activities influence bone development is not fully understood [50]. In fact, there is no consensus on the best kind of exercise to improve bone mass during childhood and adolescence and to keep bone health into adulthood.

There are many studies on the effects of physical activity in children and adolescents.

In a randomized, controlled study on children selected between first and fifth grade classes, Meyer et al. showed that the positive effects of nine months’ daily physical education program on bone mineral content (BMC) of total body, femoral neck and hip were moderately conserved over three years, regardless of the pubertal stage [51]. Jumping activities in the prepubertal years may increase PBM at the hip and lumbar spine [52]. Moderate physical activities supporting by body weight, such as running and jumping, have a more positive effect on bone accrual than activities that do not require support from weight, such as swimming [53]. Children who practice running, gymnastics and dance, show a significant increase of BMD of the neck of femur when compared to children who practice swimming. Similarly, a significant increase in BMD of lumbar spine and proximal femur is observed in boys who practice impact sports such as basketball, gymnastics and athletics, when compared to a group of adolescents who practice active load sports such as water polo and swimming [53]. The “mechanostat” concept assumes that changes in bone strength development result from the increasing loads imposed by greater muscle forces, which stimulate bone mineral acquisition. Physical activities that produce ground reaction forces are classified as impact load sports, such as gymnastics and running, while activities involving non-gravitational mechanical load are classified as active load sports, such as swimming. High-impact activities, particularly running, cheerleading, and gymnastics, appear to be more at risk for developing stress fractures than other sports [54]. Prepubertal girls who practice recreational gymnastics initiated during childhood have better bone mineral gain at the total body, lumbar spine, and forearm over 24 months. Higher-level training promotes additional gains in the forearm bone area [55]. Femoral head strength increases more with active load sport, such as swimming, with greater increase occurring during puberty. Thus, pursuing an exercise program for only a few minutes each day enhances bone mass at the proximal femur in early pubertal children [56].

There is a strong association between bone and muscle development in children, suggesting that increasing muscle mass during growth stimulates bone accrual [57]. Tennis players showed a marked increase in bone thickness on the site of muscles in the playing limb [53,57]. Similarly, racquet sport athletes demonstrated an enhanced bone mass and strength in the playing arm compared with the non-playing arm [58]. The model of “racquet sport” players explains the skeletal advantage of practicing exercise in adolescence. Racquet sport, like tennis, before puberty is associated with increased lean mass and bone mass, due to an enhanced bone size and areal BMD in the playing arm [59]. Osteogenic sports, such as football, augment BMC at the loaded sites of the skeleton, whereas the “non-osteogenic sports”, such as swimming and cycling, seem to have lower impact on BMC respect to sedentary controls [60]. This observation suggests that adolescents engaged in non-osteogenic sports should combine their exercise with weight-bearing activities in order to optimize bone development [60,61]. Individuals who practice high-impact physical activities had enhanced bone stiffness index (SI) values compared to those who practice low-impact activities or did not regularly practice exercise [62]. Thus, starting a regular physical activity, particularly during adolescence, is the key to achieving healthy bones and reducing the incidence of osteoporosis and future risk of fracture [63].

## 5. Conclusions

Bone remodeling is an essential physiological process that renews the skeleton in response to mechanical stimuli. Childhood and adolescence are critical periods in which the skeleton is most responsive to exercise. Several evidences demonstrated that the promotion of physical exercise during bone development maximizes the chances of accruing bone, potentially delaying the onset of osteoporosis in later life. The response of bone tissue to mechanical stimuli is influenced by age, hormone levels, and other metabolic factors; furthermore, it depends on the age at start, magnitude, duration and frequency of stimuli. Since the benefits of exercise are widely recognized, physical activity is considered the best non-pharmacological treatment for pathologies such as osteoporosis, obesity, diabetes and cardiovascular disease. Thus, encouraging people to be active throughout their lifespan may offer many benefits over and above bone health.

## Figures and Tables

**Figure 1 ijerph-17-01862-f001:**
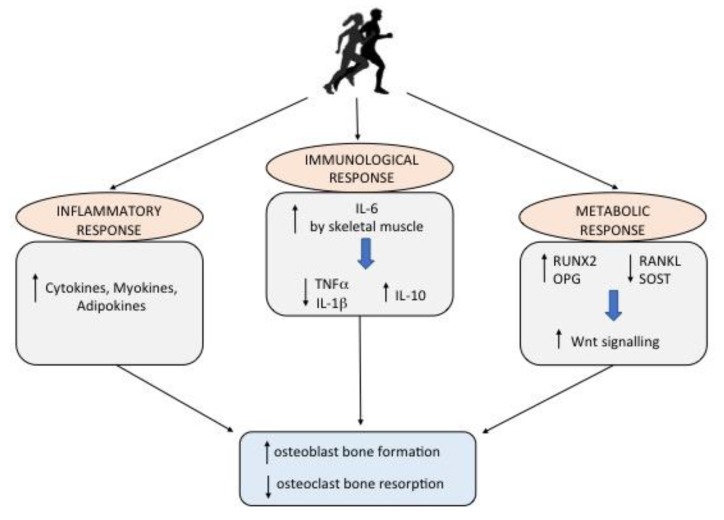
The impact of physical activity on bone health. Physical activity positively affects bone metabolism via different mechanisms: 1. activation of an inflammatory cascade involving cells of the innate and adaptive immunity and mediators of inflammation; 2. triggering an immunological response due to the increase of IL-6 by skeletal muscle; 3. stimulation of the Wnt signaling pathway.

**Figure 2 ijerph-17-01862-f002:**
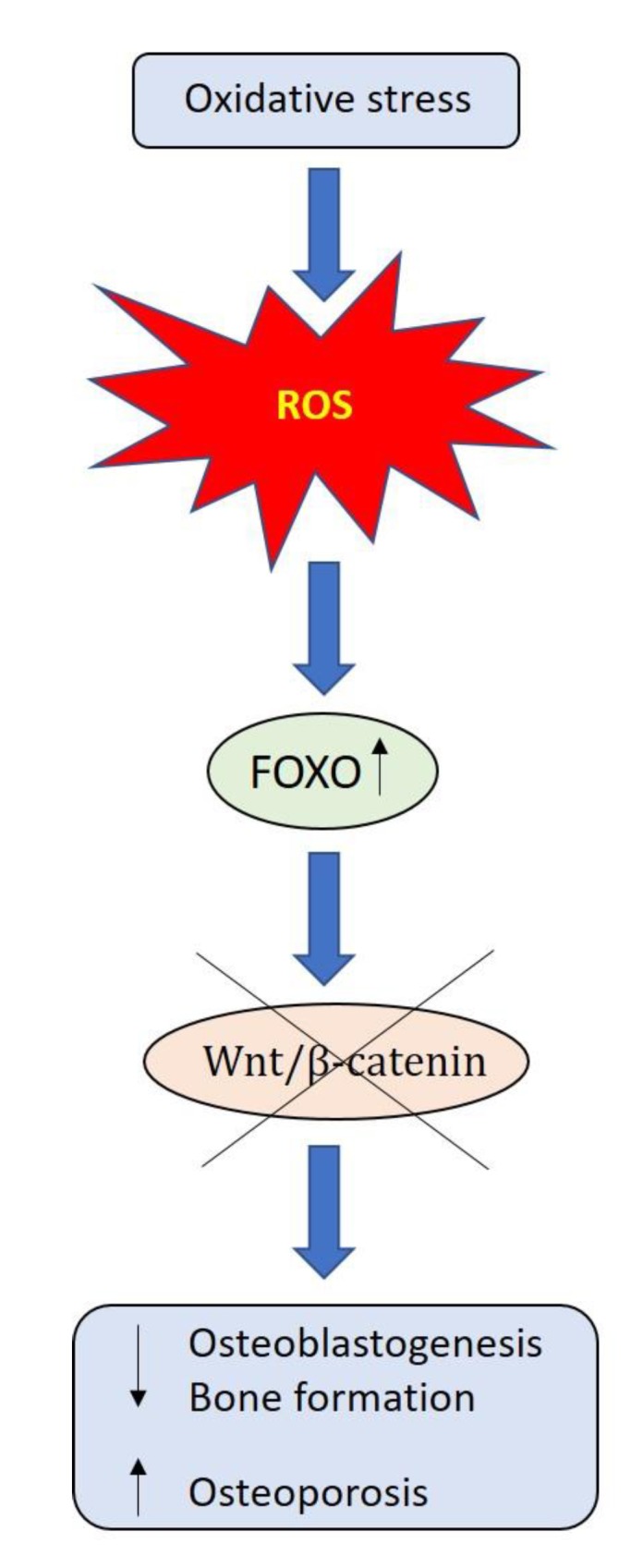
Oxidative stress determines the bone loss associated with bone ageing. The reactive oxygen species (ROS)-activated FOXO (Forkhead box O) transcription divert β-catenin away from Wnt signaling pathway, leading to decreased osteoblastogenesis.

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
