# Peer review of "How Physical Activity across the Lifespan Can Reduce the Impact of Bone Ageing: A Literature Review"

_ijerph, 2020, doi:10.3390/ijerph17061862_

Round 1

Reviewer 1 Report

Thank you for the opportunity to review your manuscript. My take home message from reading the paper was that early physical activity engagement offered some benefits and remote protective effects for bone health.

The English writing is generally sound but there are a few lapses - I marked up a PDF with some of those (apologies for the mess) and attach that for your information.

Overall it was clear that the authors are well versed in bone metabolism and the various signals that promote optimal bone accrual and protection. I enjoyed the clear explanations. I felt that the manuscript title was misleading - information on children and activity did not appear until l 150-192, with much more attention to the physiology and mechanisms  of bone turnover. I wonder whether physical activity across the lifespan would be a more accurate title? I wondered whether the links between activities, mechanical loading  and bone health could be explained in more detail. For example explaining the apparent discrepancies between  what you have termed 'active - load' sports and racquet sports. Explain what is meant/mechanisms of 'negative' impact on bone health.  I also noted that there was limited discussion of bone mineralisation and bone organisation/architecture.

I wonder whether rather than 'starting' PA during adolescence that individuals are encouraged to be active throughout their lifespan - and perhaps acknowledge that PA offers many benefits over and above bone health. 

Author Response

Reviewer 1

Thank you for the opportunity to review your manuscript. My take home message from reading the paper was that early physical activity engagement offered some benefits and remote protective effects for bone health.

The English writing is generally sound but there are a few lapses - I marked up a PDF with some of those (apologies for the mess) and attach that for your information.

Overall it was clear that the authors are well versed in bone metabolism and the various signals that promote optimal bone accrual and protection. I enjoyed the clear explanations.

  1. I felt that the manuscript title was misleading - information on children and activity did not appear until l 150-192, with much more attention to the physiology and mechanisms  of bone turnover. I wonder whether physical activity across the lifespan would be a more accurate title?

Answer 1. We really thank to the reviewer for his/her suggestion. We changed the title as suggested.

  1. I wondered whether the links between activities, mechanical loading  and bone health could be explained in more detail. For example explaining the apparent discrepancies between  what you have termed 'active - load' sports and racquet sports

Answer 2. Thank you for your observation. We explained in more details the effects of mechanical load on bone health and the differences between active, load or racquet sport.

  1. Explain what is meant/mechanisms of 'negative' impact on bone health.  

Answer 3. We rephrased the period explaining the differences between load and impact sport.

  1. I also noted that there was limited discussion of bone mineralisation and bone organisation/architecture.

Answer 5. We deeply explained the physiological mechanisms of bone remodelling. The

further addition of data on bone architecture could mislead the focus of the review.

  1. I wonder whether rather than 'starting' PA during adolescence that individuals are encouraged to be active throughout their lifespan - and perhaps acknowledge that PA offers many benefits over and above bone health. 

Answer 5. We added this suggestion in the conclusions.

Reviewer 1

Thank you for the opportunity to review your manuscript. My take home message from reading the paper was that early physical activity engagement offered some benefits and remote protective effects for bone health.

The English writing is generally sound but there are a few lapses - I marked up a PDF with some of those (apologies for the mess) and attach that for your information.

Overall it was clear that the authors are well versed in bone metabolism and the various signals that promote optimal bone accrual and protection. I enjoyed the clear explanations.

  1. I felt that the manuscript title was misleading - information on children and activity did not appear until l 150-192, with much more attention to the physiology and mechanisms  of bone turnover. I wonder whether physical activity across the lifespan would be a more accurate title?

Answer 1. We really thank to the reviewer for his/her suggestion. We changed the title as suggested.

  1. I wondered whether the links between activities, mechanical loading  and bone health could be explained in more detail. For example explaining the apparent discrepancies between  what you have termed 'active - load' sports and racquet sports

Answer 2. Thank you for your observation. We explained in more details the effects of mechanical load on bone health and the differences between active, load or racquet sport.

  1. Explain what is meant/mechanisms of 'negative' impact on bone health.  

Answer 3. We rephrased the period explaining the differences between load and impact sport.

  1. I also noted that there was limited discussion of bone mineralisation and bone organisation/architecture.

Answer 5. We deeply explained the physiological mechanisms of bone remodelling. The

further addition of data on bone architecture could mislead the focus of the review.

  1. I wonder whether rather than 'starting' PA during adolescence that individuals are encouraged to be active throughout their lifespan - and perhaps acknowledge that PA offers many benefits over and above bone health. 

Answer 5. We added this suggestion in the conclusions.

Reviewer 1

Thank you for the opportunity to review your manuscript. My take home message from reading the paper was that early physical activity engagement offered some benefits and remote protective effects for bone health.

The English writing is generally sound but there are a few lapses - I marked up a PDF with some of those (apologies for the mess) and attach that for your information.

Overall it was clear that the authors are well versed in bone metabolism and the various signals that promote optimal bone accrual and protection. I enjoyed the clear explanations.

  1. I felt that the manuscript title was misleading - information on children and activity did not appear until l 150-192, with much more attention to the physiology and mechanisms  of bone turnover. I wonder whether physical activity across the lifespan would be a more accurate title?

Answer 1. We really thank to the reviewer for his/her suggestion. We changed the title as suggested.

  1. I wondered whether the links between activities, mechanical loading  and bone health could be explained in more detail. For example explaining the apparent discrepancies between  what you have termed 'active - load' sports and racquet sports

Answer 2. Thank you for your observation. We explained in more details the effects of mechanical load on bone health and the differences between active, load or racquet sport.

  1. Explain what is meant/mechanisms of 'negative' impact on bone health.  

Answer 3. We rephrased the period explaining the differences between load and impact sport.

  1. I also noted that there was limited discussion of bone mineralisation and bone organisation/architecture.

Answer 5. We deeply explained the physiological mechanisms of bone remodelling. The

further addition of data on bone architecture could mislead the focus of the review.

  1. I wonder whether rather than 'starting' PA during adolescence that individuals are encouraged to be active throughout their lifespan - and perhaps acknowledge that PA offers many benefits over and above bone health. 

Answer 5. We added this suggestion in the conclusions.

Reviewer 2 Report

The autors report from the litterature the different mechanisms by which physical activity in childhood can reduce the impact of bone ageing.

I have some questions and comments for the authors to consider.

  • The authors should indicate how they proceeded for selecting articles for the review.
  • It lacks references on the longterm effect of physical activity during childhood. Indeed, if the beneficial effect of physical activity is well documented during lifespan, the reduction of the bone ageing with physical activity during childhood is far to being proven.
  • 1123 Does it concern physical activity during childhood or during bone ageing?

Minor corrections

Figure 2: the direction of arrows beside bone formation and resorption should be checked. In the legend, it lacks “of” after activation and stimulation

Author Response

Reviewer 2

The authors report from the litterature the different mechanisms by which physical activity in childhood can reduce the impact of bone ageing.

I have some questions and comments for the authors to consider.

1.The authors should indicate how they proceeded for selecting articles for the review.

Answer 1. We added this information in the introduction section.

  1. It lacks references on the longterm effect of physical activity during childhood. Indeed, if the beneficial effect of physical activity is well documented during lifespan, the reduction of the bone ageing with physical activity during childhood is far to being proven.

Answer 2. We really thank to the referee for his/her important observation. We added some reference (43,44) to explain this point.

  1. 1123 Does it concern physical activity during childhood or during bone ageing?

Answer 3. We think the referee refers to Foxo gene which is implicated both in physical activity and bone ageing.

Minor corrections

Figure 2: the direction of arrows beside bone formation and resorption should be checked. In the legend, it lacks “of” after activation and stimulation

Answer 2. We checked figure 2 and corrected the figure legend.

Round 2

Reviewer 2 Report

The authors have satisfactorily answered to all questions and made the necessary changes in the manuscript.